# Development of automatic generation system for lung nodule finding descriptions

Yohei Momoki[1][☯]*, Akimichi Ichinose[1][☯], Keigo Nakamura[1], Shingo Iwano[2], Shinichiro Kamiya[2], Keiichiro Yamada[2], Shinji Naganawa[2]

1 Medical Systems Research & Development Center, FUJIFILM Corporation, Minato, Tokyo, Japan,
2 Department of Radiology, Nagoya University Graduate School of Medicine, Nagoya, Aichi, Japan

☯ These authors contributed equally to this work.
* yohei.momoki@fujifilm.com

**Editor:** Vittorio Aprile, University of Pisa Department of Surgical Pathology Molecular Medicine of the Critical Area: Universita degli Studi di Pisa Dipartimento di Patologia Chirurgica Medica Molecolare e dell'Area Critica, ITALY

## Abstract

Worldwide, lung cancer is the leading cause of cancer-related deaths. To manage lung nodules, radiologists observe computed tomography images, review various imaging findings, and record these in radiology reports. The report contents should be of high quality and uniform regardless of the radiologist. Here, we propose an artificial intelligence system that automatically generates descriptions related to lung nodules in computed tomography images. Our system consists of an image recognition method for extracting contents–namely, bronchopulmonary segments and nodule characteristics from images–and a natural language processing method to generate fluent descriptions. To verify our system's clinical usefulness, we conducted an experiment in which two radiologists created nodule descriptions of findings using our system. Through our system, the similarity of the described contents between the two radiologists ($p = 0.001$) and the comprehensiveness of the contents ($p = 0.025$) improved, while the accuracy did not significantly deteriorate ($p = 0.484$).

## Introduction

Lung cancer is the most common and leading cause of cancer-related deaths, worldwide [1]. Computed tomography (CT) is useful for early detection and diagnosis of lung cancer [2–4]. Regarding lung cancer diagnoses, many studies have compared CT and pathological findings, and the histological background of the marginal and internal characteristics of lung nodules is well understood [5,6]. Based on these studies, several imaging findings have been identified that affect the differentiation of benign and malignant nodules and their management [2,7,8].

To communicate the radiographic interpretation results to physicians, radiologists prepare a radiology report that summarizes the imaging findings and diagnoses. Regarding lung nodules, information that affects benign/malignant discrimination (e.g., anatomical location, size, marginal/internal characteristics, and relationship with surrounding structures), should be accurately described in the report. Thin-section CT is recommended for accurate characterization and measurement [2]. This implies that the radiologist needs to identify many slices and record a variety of imaging findings, which takes a long time. Furthermore, it is desirable that

**Data Availability Statement:** The CT images used in the observer performance test are available from https://www.cancerimagingarchive.net/collection/rider-lung-ct/ Other relevant data are within the manuscript and its Supporting Information files.

**Funding:** This research is a cooperative study between FUJIFILM Corporation and Nagoya University Graduate School of Medicine. IS received a specific grant (2618Dn-02b) from FUJIFILM Corporation. The funders and IS analyzed the collected data together. The funder had roles in study design, data collection, software development, analysis, decision to publish, and manuscript preparation. No authors received personal support from the funder.

**Competing interests:** I have read the journal's policy and the authors of this manuscript have the following competing interests: This research is a cooperative study between FUJIFILM Corporation and Nagoya University Graduate School of Medicine. This does not alter our adherence to PLOS ONE policies on sharing data and materials.

the content described in the report is of high quality and uniform; however, at present, achieving this varies depending on the radiologist. Furthermore, as mentioned, there is a variety of content to be verified. Hence, the risk of omission of important content exists. By developing an artificial intelligence (AI) system that supports the creation of finding descriptions, it is expected that the quality and uniformity of reports will be improved, and the burden of interpretation work will be reduced.

In recent years, many diagnostic support systems using deep learning (DL) techniques, such as convolutional neural networks and recurrent neural networks, have been studied; however, most of these studies focused on lesion detection and differentiation [9,10], and few focused on report generation. An end-to-end method has been proposed to generate finding descriptions directly from chest X-ray images [11,12]. These studies used hundreds of thousands of image/findings pairs for training. To automatically generate finding descriptions of lung nodules on CT images, it is necessary to comprehensively extract detailed imaging findings related to nodules from three-dimensional (3D) images. Compared with the finding descriptions of the X-ray image, the finding descriptions of the CT image exhibit a large individual difference in the content to be described, and it can be assumed that the same approach requires a large amount of training data to absorb this individual difference.

This study's aim is to develop a DL-based system that generates the descriptions of findings from lung nodules on CT images. To achieve a system with clinically practical accuracy using a realistically collectable amount of training dataset, we will develop the system in three modules: (1) bronchopulmonary segments prediction AI, (2) nodule classification AI, and (3) finding description generation AI. First, we evaluate whether the accuracy of the bronchopulmonary segments prediction AI and the nodule classification AI are compatible with those of the two radiologists, because these AIs are responsible for extracting the contents to be described in the findings from CT images with sufficient accuracy. Second, we evaluate the adequacy and fluency of the finding description generation AI that is responsible for realizing fluent descriptions, including sufficient content generated by the above AIs. Third, we evaluate the clinical usefulness of the system by comparing the uniformity of contents among two radiologists and the comprehensiveness of the contents in the case where the radiologists created finding descriptions from drafts generated by the system to the case where they created drafts from scratch.

## Materials and methods

This study was approved by the Ethics Review Committee of Nagoya University Graduate School of Medicine and Nagoya University Hospital (the approval number is #2017–0471) and the Bioethics Review Committee of FUJIFILM Corporation (the approval number is #104). Both ethics committees waived the requirement for patient consent. All data were analyzed anonymously and all experiments were performed in accordance with the relevant guidelines and regulations.

### Clinical data

To develop the algorithm, we collected radiology reports and chest CT images of 9,826 and 4,100 patients from one and two facilities in Japan, respectively. The data was collected from April 1, 2006, to March 31, 2016 by the facilities. We analyzed the reports to identify examinations in which at least one lung nodule was detected. Based on the results, images and reports were collected without considering the oncological history or examination purpose (screening, preoperative evaluation, etc.). All data were collected anonymously, and the authors had no access to any information that could identify individual participants during or after the data

**Table 1. Specification of dataset.**

| Data Type | Characteristic | # |
|---|---|---|
| Report | No. of reports | 9,826 |
| | No. of finding descriptions | 11,025 |
| | No. of patients | 5,953 |
| | • No. of men | 3,661 |
| | • No. of women | 2,297 |
| | Mean age | 67.6+/- 11.9 |
| Image | No. of chest CTs | 4,623 |
| | • No. of CT/report pairs | 3,224 |
| | No. of patients | 3,526 |
| | • No. of men* | 1,934 |
| | • No. of women* | 1,244 |
| | Mean age* | 68.3 +/- 11.8 |
| | Reconstruction | |
| | • Lung Kernel | 4,109 |
| | • Abdomen Kernel | 514 |
| | Slice Thickness | |
| | • < = 1.0 mm | 2,698 |
| | • < = 2.0 mm | 1,094 |
| | • < = 3.0 mm | 237 |
| | • < = 4.0 mm | 1 |
| | • < = 5.0 mm | 593 |

In some of public data, sex and age are anonymized; therefore, these data are excluded from the tabulation of the items marked with *.

collection process. These data were assessed for research purposes from February 14, 2018, to March 31, 2020. Furthermore, we used 523 cases from the lung image database consortium image collection (LIDC-IDRI) [13], which is available from The Cancer Imaging Archive under the Creative Commons Attribution 3.0 Unsupported License [14]. The details of the dataset are summarized in Table 1.

## Proposed framework overview

We developed a DL-based system, which automatically generated lung nodule finding descriptions using the inputted chest CT images and coordinates of the lung nodule region. As shown in Fig 1, the system comprised three modules: (1) bronchopulmonary segment prediction, (2) nodule classification, and (3) description generation. Each module was trained separately.

## Bronchopulmonary segments prediction

The lungs can be subdivided into five lobes and 18 segments. In these findings, the lobes/segments were commonly mentioned for easy location of the nodule. This information is crucial for accurate diagnosis and treatment planning. This module consisted of the lung lobe and bronchopulmonary segment segmentation modules.

**Lung lobe segmentation module.** In this module, the lung field was divided into five lobes using the U-Net architecture [15] (see S1 Fig). This architecture has been widely used in various medical image segmentation tasks and has is highly accurate [16,17]. For training and validation, 226 and 20 data points, respectively, were extracted from the development dataset,

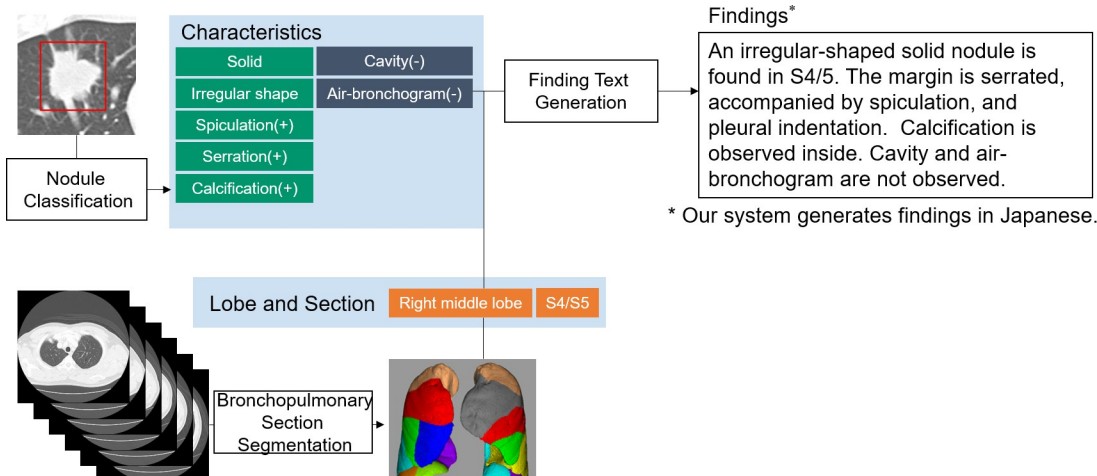

**Fig 1. Overview of the proposed system.** Anatomical information and characteristics of the target nodules are extracted from CT images using image recognition modules. Text generation module integrates them to generate finding sentences. Reprinted under a CC BY license, with permission from Shingo Iwano, original copyright 2024.

and ground truth data were prepared by medical professionals under the supervision of board-certified radiologists. As a preprocessing step, the image spacing was resized to 1 mm/voxel, and the CT values were clipped from −1,400 to 200 and linearly normalized to obtain a value of 0–1. We trained the network with a batch size of two with 100,000 steps. Dice loss was used as the loss function [18] and optimized using the Adam optimizer. The initial learning rate of $10^{-3}$ was reduced by a factor of 0.1 at 50,000 steps. Data augmentation techniques, such as sharpness transformation, scaling, and random cropping, were used. All modules, including lobe segmentation, were implemented using TensorFlow [19].

**Bronchopulmonary segmentation module.** This module divided each lobe region into bronchopulmonary segments. Similar to lung lobe segmentation module, the U-Net architecture was used for this module. Preparing ground truth data with clear segment boundaries ("densely labeled data") was difficult as the boundaries of each segment could not be observed on the CT images; therefore, we proposed a training approach using localized ground truth data obtained through radiology report analysis (hereafter referred to as "sparsely labeled data").

Fig 2 shows an example of a bronchopulmonary segment with a nodule and its description in a the radiology report. The reports also included annotations from radiologists, such as arrows pointing to the lesion. We combined this information to identify the segment name of the area surrounding the lesion. Out of a total of 734 created cases, 714 and 20 cases were used for training and validation, respectively.

However, our approach exhibited decreased accuracy in segmenting boundaries compared to the use of densely labeled data. This is because we did not have access to information about class boundaries; and training the network about the boundaries of the bronchopulmonary segments was difficult. To address this, we used the linear nature of boundaries and introduced an additional loss function called "edge loss" into the general segmentation loss function (Dice loss) by extending the method by Javanmardi et al.[20].

$$L_{seg} = \sum_{n=1}^{N} \sum_{x_n \in I_n} |\nabla h(f(x_n))| \qquad (1)$$

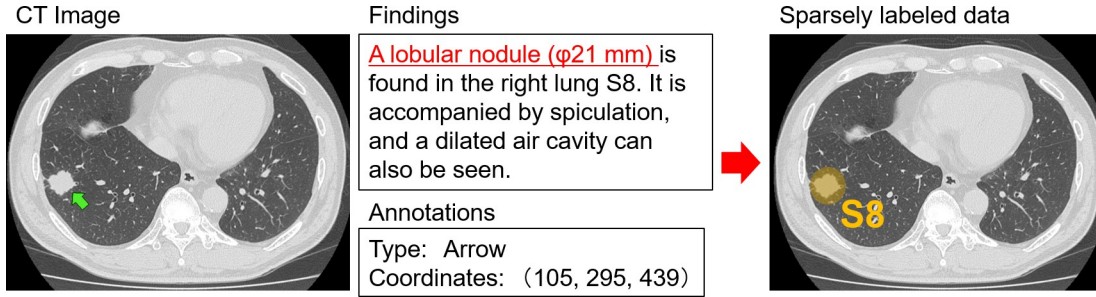

**Fig 2. Creation of sparsely labeled data.** By analyzing the findings, the bronchopulmonary segments, where the lesion is located, can be found. Also, the coordinate information of the lesion is recorded as annotation data. This information is combined to create localized ground truth data of bronchopulmonary segments. Reprinted under a CC BY license, with permission from Shingo Iwano, original copyright 2024.

$$h(x) = \frac{e^{\frac{x}{T}}}{\sum_{c=1}^{C} e^{\frac{x_c}{T}}} \tag{2}$$

where *f* is the network, *N* the batch size, $I_n$ the voxel set of the *n*-th image in the mini-batch, *C* is the number of classes, and *T* is the temperature value of the softmax function *h*, which was set to 0.5 in our experiment. We processed the network's output using a Sobel filter as a differential operation to smoothen the class boundaries by reducing the fluctuations in the probability map. The preprocessing method, learning rate, optimizer, and data augmentation were the same as those used for lung lobe segmentation.

*Nodule classification.* This module was developed to automatically extract the imaging findings required by radiologists to diagnose lung nodules. These findings were determined based on the diagnostic imaging guidelines for lung cancer [2,7], Lung-RADS [21], and shape classification of lung nodules [22]. In total, we defined 14 items as the target findings, as detailed in Table 2.

We used a CNN-based model, which is a simpler version of the VGG [23] (see S2 Fig), with 12 3D-convolutional layers and two max pooling layers. Owing to the small amount of data

**Table 2. List of image findings of lung nodules.**

| Category | Class |
| --- | --- |
| Opacity | solid/part-solid/pure ground-glass type |
| Shape | irregular/round shape |
| Margin | ill/well-defined |
| | irregular/smooth edge |
| | spiculated(+)/(-) |
| | lobulated(+)/(-) |
| | ragged(+)/(-) |
| | polygonal(+)/(-) |
| Internal Characteristics | air-bronchogram(+)/(-) |
| | cavity(+)/(-) |
| | calcification(+)/(-) |
| | fat(+)/(-) |
| External Characteristics | pleural contact(+)/(-) |
| | pleural indentation(+)/(-) |

(4,500), the output of the last convolution layer was compressed to 256 dimensions by global average pooling to avoid overfitting. The high interobserver variability in some classes obstructed the creation of consistent ground-truth data. To handle these inconsistent and noisy training data, we adopted the bootstrapping sigmoid cross-entropy loss [24].

Among the target classes, some classes that could be determined with completely different features, including fine shadows inside or around the nodule (ragged, air-bronchogram, etc.) or the relationship with the surroundings (pleural contact/indentation, etc.). In such cases, problem solving using separate networks is common. However, in this case, capturing the relationships between classes was not possible. Therefore, we trained a network that inputted a wide-area and enlarged images together in the channel direction. The wide-area image was a CT image was rescaled to 1 mm and clipped to a margin of 15 mm in the nodule region. The enlarged image was a clip of the nodule region and rescaled to the same size as the wide-area image.

Ground truth data from 4,197 series (4,952 nodules) for the training and 66 series (66 nodules) for validation were created by two board-certified radiologists. The ground truth data creation criteria were discussed in advance. As a rule, each case was observed by one reader, and only the cases in which determining the characteristics was difficult were checked by the other reader. As a preprocessing step, we cropped 3D nodule patches from the CT images. The CT values were clipped from −1,400 to 200 and linearly normalized to assume a value of 0–1. Training was performed with a batch size of 12. The initial learning rate of $10^{-4}$, which was multiplied by 0.1 in 100,000 steps. The AdamW optimizer was used. The data augmentation transformations included sharpness transformations, scaling, and random cropping.

## Lung nodule description generation

This module output fluent sentences that include only sufficient input content, such as bronchopulmonary segments and nodule characteristics. However, the original nodule descriptions in actual clinical reports also included various information such as the name of the disease, comparative findings, and dates. Therefore, if we train the description-generation model using the original text as an output, the model will generate hallucinate phrases. To avoid hallucinations, we edited the original text using a manual annotation process, which included:

1. Lung nodule/mass descriptions were extracted from the findings section of radiology reports.

2. Words other than the characteristics, bronchopulmonary segments, and major axis of the nodule/mass were deleted from the descriptions.

3. The anatomical tissue names, nodule/mass, and axis measurements were replaced from in the description with special tokens. Note that these tokens were to be replaced with appropriate values after the description generation.

4. Terms were uniformed to follow the latest Japanese lung cancer guideline [7].

5. Grammatical errors induced by the previous steps were corrected.

6. The following training processes were repeated several times to increase the robustness of the model: training the model with the created dataset, generating outputs with the unknown label combinations, validating the output texts, and appending them to the dataset.

We adopted a sequence-to-sequence model [25] with a general attention mechanism [26] incorporating long short-term memory.

## Evaluation of bronchopulmonary segments prediction / nodule classification performance

We selected 100 cases as the evaluation dataset for the bronchopulmonary segment prediction and nodule classification modules. This dataset consisted of cases selected to cover the characteristics and positions of the nodules based on the descriptions in radiology reports. For objective evaluation, patients from the training dataset were not included in the evaluation dataset. At least 10 positive/negative cases were collected for each class (fat was excluded because nodules with internal fat appeared infrequently, and only three positive cases could be collected). For the 100 collected cases, a 3D rectangle was attached to the lung nodule, and one researcher (with more than 26 years of experience in chest radiological interpretation) recorded the 14 characteristics and segment names of the nodule. When multiple nodules were found in the lungs, a radiologist selected one representative nodule as the evaluation target. Certain characteristics could not be determined depending on the case. In these cases, only the characteristics that could be determined were evaluated. Table 3 and Fig 3 present the details of the evaluation data.

For bronchopulmonary segment prediction, the exact/partial match rate was calculated by comparing the prediction results with the gold standard. For cases in which the lung nodules spanned multiple segments, segments that occupied $\geq 10\%$ of the volume of the rectangle were regarded as the prediction results. Furthermore, to verify the effect of the aforementioned edge loss, the performance of models with and without edge loss was compared. To reduce the influence of differences in the initial values of the parameters, experiments were repeated five times for each condition. The required number of trials was calculated by *a priori* power analysis ($\alpha = 0.05$, $\beta = 0.80$; effect size was calculated from prior experiments). Two radiologists with four years of radiological interpretation experience also identified the segments and compared their performance with that of our module.

For nodule classification, we selected area under the curve (AUC) as an evaluation metric, which was the area under the receiver operating characteristic (ROC) curve. To verify the effect of the enlarged image input, the performances of the models that input both the enlarged and wide-area images and only the wide-area image were compared. To reduce the influence of differences in the initial values of the parameters, experiments were repeated seven times

**Table 3. Specification of evaluation dataset.**

| Characteristic | # |
|---|---|
| No. of chest CTs | 100 |
| No. of patients | 99 |
| • No. of men | 56 |
| • No. of women | 43 |
| Mean age | 64.4 +/- 13.9 |
| Slice Thickness | |
| • 0.5mm | 45 |
| • 1.0mm | 55 |
| Reconstruction | |
| • Lung Kernel | 100 |
| Lobe with nodule to be evaluated | |
| • Right Upper Lobe | 29 |
| • Right Middle Lobe | 8 |
| • Right Lower Lobe | 26 |
| • Left Upper Lobe | 25 |
| • Left Lower Lobe | 12 |

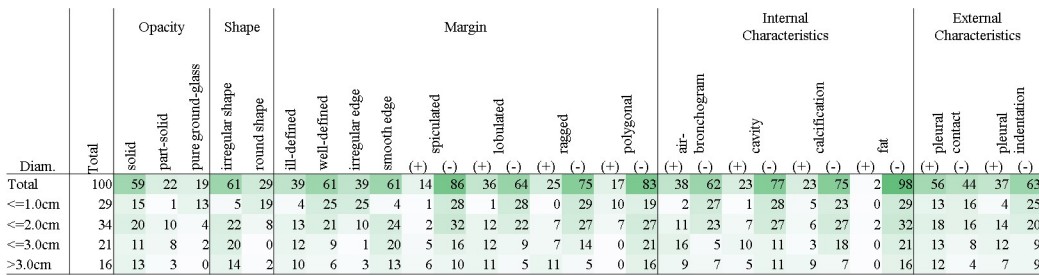

| Diam. | Total | Opacity | | | Shape | | Margin | | | | | | | | | | | | Internal Characteristics | | | | | | | | External Characteristics | | | |
|---|---|---|---|---|---|---|---|---|---|---|---|---|---|---|---|---|---|---|---|---|---|---|---|---|---|---|---|---|---|---|
| | | solid | part-solid | pure ground-glass | irregular shape | round shape | ill-defined | well-defined | irregular edge | smooth edge | spiculated (+) | (-) | lobulated (+) | (-) | ragged (+) | (-) | polygonal (+) | (-) | air-bronchogram (+) | (-) | cavity (+) | (-) | calcification (+) | (-) | fat (+) | (-) | pleural contact (+) | (-) | pleural indentation (+) | (-) |
| Total | 100 | 59 | 22 | 19 | 61 | 29 | 39 | 61 | 39 | 61 | 14 | 86 | 36 | 64 | 25 | 75 | 17 | 83 | 38 | 62 | 23 | 77 | 23 | 75 | 2 | 98 | 56 | 44 | 37 | 63 |
| <=1.0cm | 29 | 15 | 1 | 13 | 5 | 19 | 4 | 25 | 25 | 4 | 1 | 28 | 1 | 28 | 0 | 29 | 10 | 19 | 2 | 27 | 1 | 28 | 5 | 23 | 0 | 29 | 13 | 16 | 4 | 25 |
| <=2.0cm | 34 | 20 | 10 | 4 | 22 | 8 | 13 | 21 | 10 | 24 | 2 | 32 | 12 | 22 | 7 | 27 | 7 | 27 | 11 | 23 | 7 | 27 | 6 | 27 | 2 | 32 | 18 | 16 | 14 | 20 |
| <=3.0cm | 21 | 11 | 8 | 2 | 20 | 0 | 12 | 9 | 1 | 20 | 5 | 16 | 12 | 9 | 7 | 14 | 0 | 21 | 16 | 5 | 10 | 11 | 3 | 18 | 0 | 21 | 13 | 8 | 12 | 9 |
| >3.0cm | 16 | 13 | 3 | 0 | 14 | 2 | 10 | 6 | 3 | 13 | 6 | 10 | 11 | 5 | 11 | 5 | 0 | 16 | 9 | 7 | 5 | 11 | 9 | 7 | 0 | 16 | 12 | 4 | 7 | 9 |

**Fig 3. Size and characteristic distribution of lung nodules in the evaluation dataset.**

each. The required number of trials was calculated by *a priori* power analysis (α = 0.05, β = 0.80; effect size was calculated from prior experiments). Two radiologists performed the characterization and compared the performance of our module based on the ROC curve.

## Evaluation of nodule description generation

We randomly divided the dataset into training/validation/test segments to avoid duplication of the combination of input contents. The volumes were as follows: training: 25,327; validation: 735; and test: 713.

We evaluated the descriptions generated from two perspectives: adequacy and fluency. Regarding adequacy, we evaluated the number of agreements between the input and output contents using TP, FN, and FP as evaluation metrics. For fluency, we used the manual count of ungrammatical descriptions as the evaluation metric.

## Observer performance test

We used the RIDER Lung CT dataset [27] for the observer performance test, available from The Cancer Imaging Archive under the Creative Commons Attribution 3.0, Unsupported License [14]. Two radiologists with four years of clinical experience interpreted 32 nodules (See S3 Fig) in the dataset and created 32 descriptions of findings for the quantitative assessment of lung nodules in each of the following two settings:

1. Description creation from scratch without using the system (w/o AI)

2. Description creation by modifying the drafts created by the system (w/ AI).

   The following procedure is the gold standard:

1. One researcher created the descriptions from scratch.

2. Content was manually extracted from all descriptions, including those created by the subject radiologists in the above two settings.

3. The researcher created a gold standard by assigning the following four ranks to the content of each case: (A) content that must be described, (B) content that should be described, (C) any content, and (D) wrong content.

The A–C ranks were judged based on the magnitude of influence on benign/malignant differentiation, staging, and treatment method selection. Fig 4 shows the gold standard examples, and Fig 5 shows the content and rank trends of the observer performance test dataset.

For the observer performance test, we used the following two evaluation metrics: (i) agreement of content between the radiologists and (ii) accuracy and comprehensiveness of content. For metric (i), we accumulated the number of matches of the A/B content described by the subject

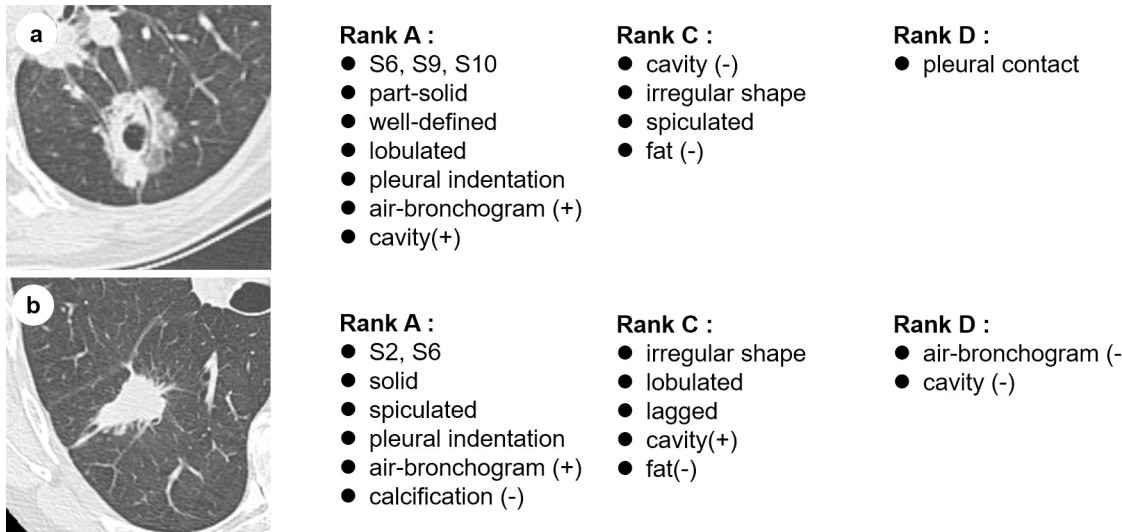

**Fig 4. The gold standard examples of the observer performance test dataset.** Reprinted from https://www.cancerimagingarchive. net/collection/rider-lung-ct/ under a CC BY license, with permission from Zhao, B., Schwartz, L. H., & Kris, M. G., original copyright 2015.

radiologists for each case. Regarding the metric (ii), we adopted the number of true positives, false negatives, and false positives as the evaluation metrics by accumulating the A/B content as TP and FN, and the D content as FP. S1 Data shows the radiologist-generated descriptions and gold standards for the nodules, and the statistical data of the observer performance test.

## Results

### Evaluation of bronchopulmonary segments prediction

Fig 6(A) and 6(B) show the evaluation results of the bronchopulmonary segment prediction, and Fig 6(C) shows an example of bronchopulmonary segment prediction. When optimizing

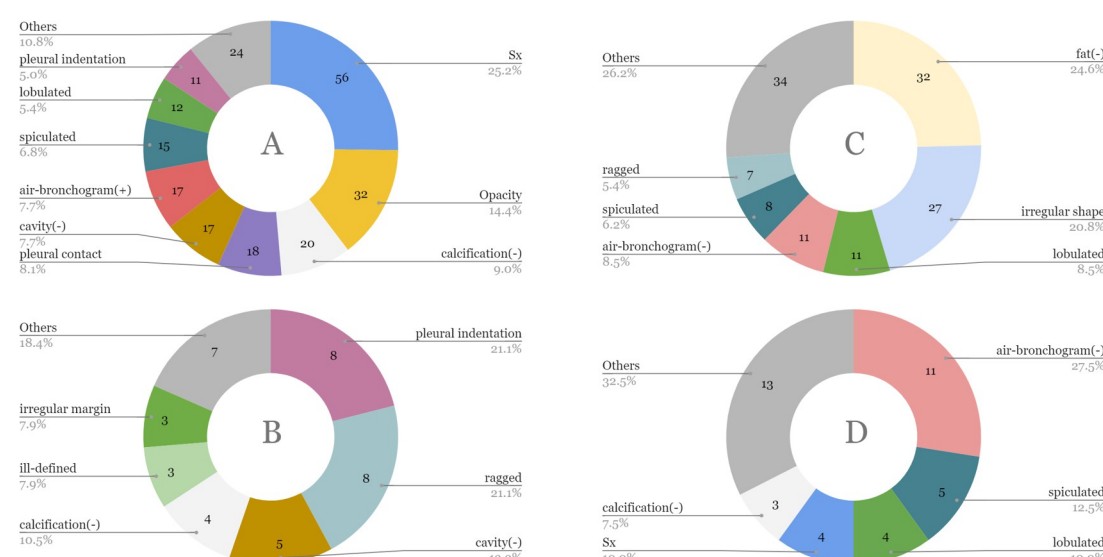

**Fig 5. Content and rank trend of the observer performance test dataset.** (A) contents that must be described, (B) contents that should be described, (C) any content, and (D) wrong content.

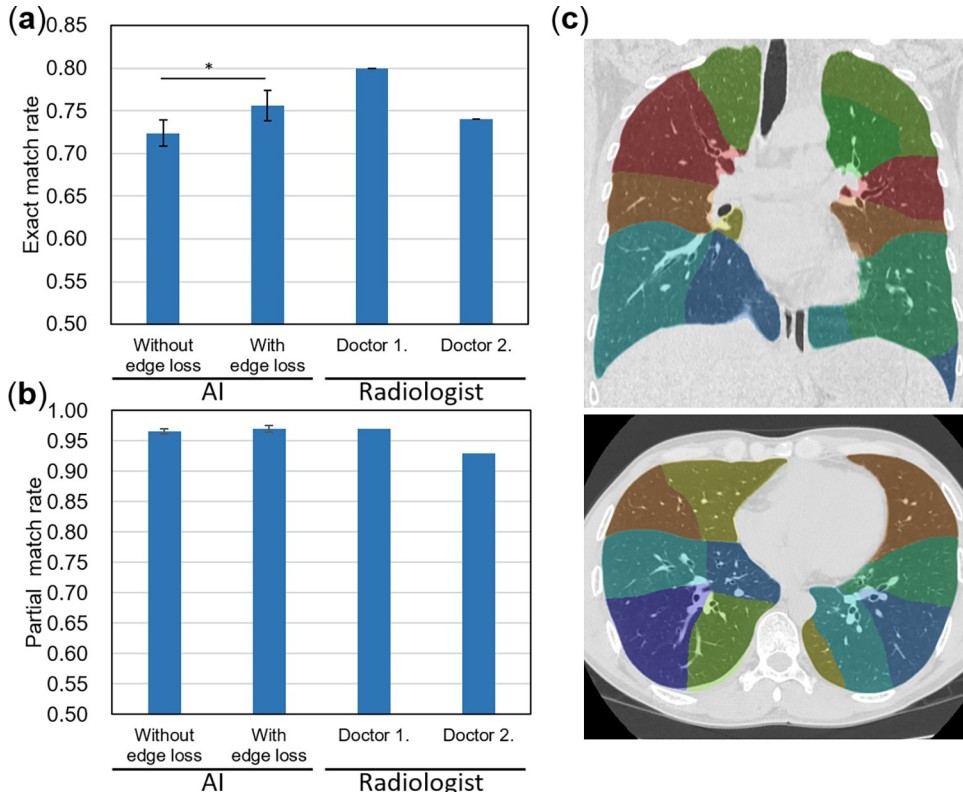

**Fig 6. Performance evaluation results and examples of bronchopulmonary segments prediction.** (a/b) Comparison of the exact match rate/partial match rate between ground truth and prediction results by AI with/without edge loss and two radiologists. * indicates p < 0.05. (c) Examples of prediction results. Different colors mean different segments. Reprinted under a CC BY license, with permission from Shingo Iwano, original copyright 2024.

the segmentation network based on edge loss to smoothen the boundaries of segments, the exact match rate between the segment identification results of nodule regions obtained by a skilled radiologist and those by AI was 75.6%, and the partial match rate 97.0%. There was an improvement in the exact match rate compared to not applying edge loss (p = 0.024; two-tailed *t*-test). Comparing the two radiologists' reports, neither the exact nor the partial match rates were significantly different from the AI performance. Table 4 shows the accuracy of the bronchopulmonary segment prediction for each lobe. There were no cases in which the estimation of the lung lobe failed.

**Table 4. Accuracy of bronchopulmonary segments prediction.**

|  | Partial Match Rate | Exact Match Rate |
|---|---|---|
| Entire Lung | 97.0% | 75.6% |
| Right Upper Lobe | 100.0% | 84.1% |
| Right Middle Lobe | 95.0% | 75.0% |
| Right Lower Lobe | 96.2% | 68.5% |
| Left Upper Lobe | 100.0% | 85.6% |
| Left Lower Lobe | 85.0% | 50.0% |

The average success rates (partial/exact match rate between gold standard and predicted results) for each lung lobe and entire lung are shown.

In 21 patients, insufficient interlobar fissures were observed in the lobes containing the target nodule. The exact match rates between the gold standard and predicted results were 71.4% and 76.7% for cases with and without insufficient interlobar fissures, respectively, (p = 0.15; two-tailed $t$-test). Additionally, five cases of lobe deformity due to lesions within the lung (pneumonia or lung cancer) were observed, and in one case, the target lung lobe was partially resected. The exact match rate for the former was 88.0%, whereas that for the latter was 60.0%.

## Evaluation of nodule classification

Fig 7(A) shows the performance when an enlarged image and a wide-area image were input, and when only a wide-area image was input. By inputting an enlarged image, the performance improved in all classes except the straight class, and there was a significant performance improvement for many of the marginal characteristic and internal characteristic classes (two-tailed $t$-test). The AUC decreased only for the straight class, but no significant difference was found (p = 0.341; two-tailed $t$-test). Fig 7(B) shows the ROC curve for each class and the class estimation results of the two radiologists. In the 17 classes, the prediction performance of AI was equivalent to that of the radiologists. In the categories related to marginal characteristics such as lobulation and ragged structure, there were several classes in which AI performed better.

In Fig 8, the classification performance at the cutoff value based on the Youden index is presented. While high performance is achieved for opacity and internal characteristics, performance is relatively poor for marginal characteristics.

## Evaluation of nodule description generation

Regarding adequacy, there were 710 (4,618) of the true positives (TP), 0 (0) of the false positives (FP), and 3 (3) of the false negatives (FN) cases (contents). Regarding fluency, the number of cases of grammatical problems was 1/710, and the error was local (one particle error).

## Observer performance test

Table 5 shows the agreement of the described contents between the two radiologists in the w/ AI setting, improved with a significant difference (p = 0.001; two-tailed paired $t$-test), compared to that in the w/o AI setting. Furthermore, the number of TP contents in the w/ AI setting improved (p = 0.025; two-tailed paired $t$-test), compared to the w/o AI setting without increasing the number of FP contents with a significant difference (p = 0.484; two-tailed paired $t$-test).

Furthermore, we revealed the contents with significant improvement by analyzing cases where the correct contents in the draft provided by the system resulted in an improvement of the radiologist's description. The contents of the top three TP improvements were:

- Bronchopulmonary segments (five cases)

- Pleural contact + (five cases)

- Pleural indentation sign + (four cases)

Fig 9 presents the examples including images and descriptions.

## Discussion

In our research, we developed a system that generated descriptions of findings from lung nodules on CT images using a realistically collectable amount of training dataset which consisted

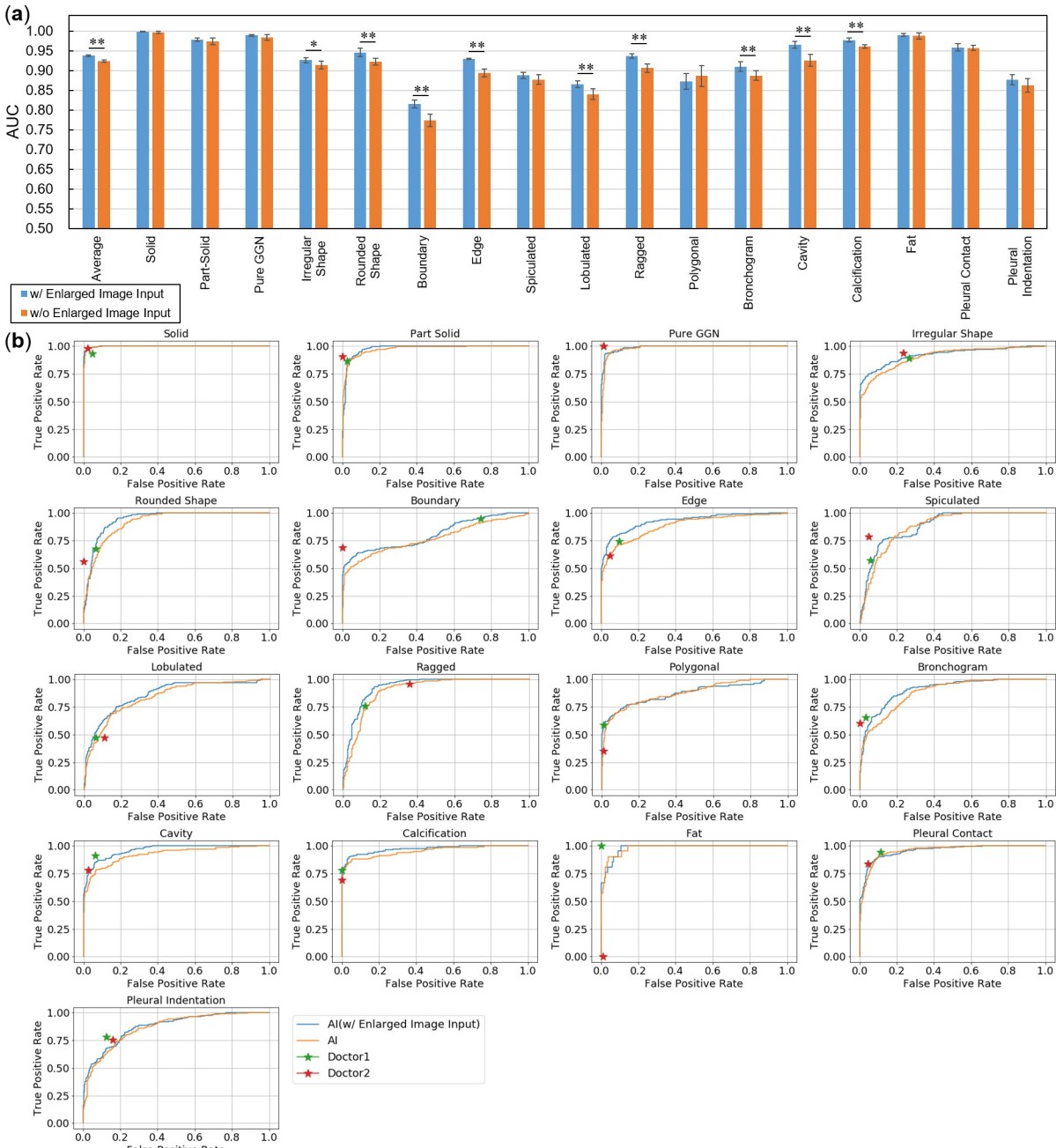

**Fig 7. Comparative evaluation results of lung nodule classification.** (a) Comparison of AUC for each class between AI with enlarged image input and one without. (b) AI and individual radiologist classification results. ** and * indicates p < 0.01 and p < 0.05, respectively.

9,800 radiology reports and 4,600 CT images. Our experiments showed that our system achieved similar accuracy as radiologists in recognizing bronchopulmonary segments and nodule characteristics, and assisted radiologists in writing lung nodule descriptions more uniformly and comprehensively without sacrificing precision.

Radiologists interpret many images, confirm numerous findings for each, and describe the contents necessary for benign/malignant discrimination in the findings section of the report in natural language format in their daily work. In particular, lung nodules in CT images have

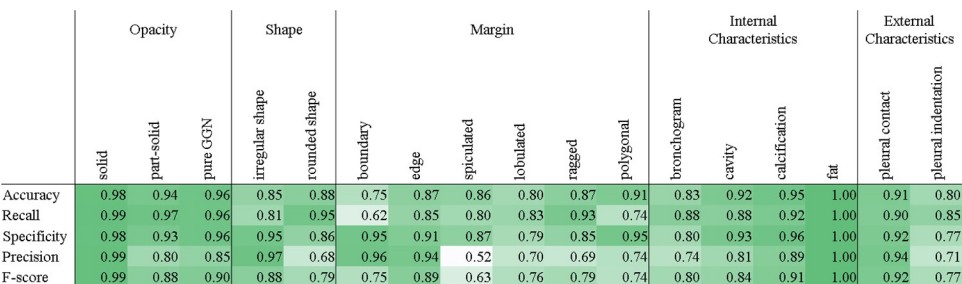

| | Opacity | | | Shape | | Margin | | | | | | Internal Characteristics | | | | External Characteristics | |
|---|---|---|---|---|---|---|---|---|---|---|---|---|---|---|---|---|---|
| | solid | part-solid | pure GGN | irregular shape | rounded shape | boundary | edge | spiculated | lobulated | ragged | polygonal | bronchogram | cavity | calcification | fat | pleural contact | pleural indentation |
| Accuracy | 0.98 | 0.94 | 0.96 | 0.85 | 0.88 | 0.75 | 0.87 | 0.86 | 0.80 | 0.87 | 0.91 | 0.83 | 0.92 | 0.95 | 1.00 | 0.91 | 0.80 |
| Recall | 0.99 | 0.97 | 0.96 | 0.81 | 0.95 | 0.62 | 0.85 | 0.80 | 0.83 | 0.93 | 0.74 | 0.88 | 0.88 | 0.92 | 1.00 | 0.90 | 0.85 |
| Specificity | 0.98 | 0.93 | 0.96 | 0.95 | 0.86 | 0.95 | 0.91 | 0.87 | 0.79 | 0.85 | 0.95 | 0.80 | 0.93 | 0.96 | 1.00 | 0.92 | 0.77 |
| Precision | 0.99 | 0.80 | 0.85 | 0.97 | 0.68 | 0.96 | 0.94 | 0.52 | 0.70 | 0.69 | 0.74 | 0.74 | 0.81 | 0.89 | 1.00 | 0.94 | 0.71 |
| F-score | 0.99 | 0.88 | 0.90 | 0.88 | 0.79 | 0.75 | 0.89 | 0.63 | 0.76 | 0.79 | 0.74 | 0.80 | 0.84 | 0.91 | 1.00 | 0.92 | 0.77 |

**Fig 8. Performance evaluation results of lung nodule classification.**

various characteristics to distinguish benign/malignant, so the ways in which radiologists describe their reports tend to be diverse. The Japanese lung cancer guidelines require radiologists to perform independent double interpretations and comparative interpretations of past CT images [7]. The more diverse the descriptions in reports are for each radiologist, the less efficient their work would be, because it is quite common for them to read radiology reports described by other radiologists. Thus, improving the comprehensiveness and accuracy of contents in report descriptions and making them uniform among radiologists are significant problems in this field. Our approach mitigated this problem by providing radiologists with an accurate and uniform draft of the lung nodule descriptions. We believed that our approach benefitted both radiologists and patients by improving not only the work efficiency and quality of radiologists, but also promoting radiomics research that combined radiology reports and images by improving the searchability of reports.

Our system comprised three modules to achieve clinically practical performance using a realistically collectable number of training datasets: bronchopulmonary segment prediction, nodule classification, and description generation. This was a point that differed from previous studies [11,12].

There are few previous studies on bronchopulmonary segment prediction, although several studies have been conducted on lung lobe segmentation [28,29]. Rikxoort et al. proposed a bronchopulmonary segment prediction method based on the distance from the interlobular pleuras [30]. Their method did not use bronchi and blood vessel information, so it did not reflect individual differences in bronchial bifurcations. Our method adopted a DL model trained with the data created based on bronchial information; thus, our method could reflect individual differences in bronchial bifurcations. In our experiments, our model achieved similar accuracy as the radiologists. The proposed edge loss led to a significant improvement in the exact match rate. In contrast, the model trained without edge loss resulted in unstable segment boundaries and misrecognition of large nodules spanning multiple bronchopulmonary

**Table 5. The result of observer performance test.**

| Metric | Condition | Mean ± Standard Deviation | P value |
|---|---|---|---|
| Agreed contents | w/o | 4.81 ± 1.62 | 0.001* |
| | w/ | 5.63 ± 1.54 | |
| True positive contents | w/o | 5.88 ± 1.62 | 0.025* |
| | w/ | 6.20 ± 1.65 | |
| False positive contents | w/o | 0.34 ± 0.57 | 0.484 |
| | w/ | 0.41 ± 0.63 | |

*statistically significant.

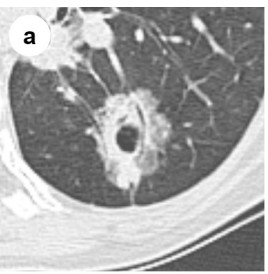

**Generated description**
An irregular, part-solid mass with major axis of 36 mm is found
in contact with pleura in the lower lobe S6/S9/S10 of the left lung.
It has lobulated and spiculated margins with pleural indentation.
Air-bronchogram and cavity are found inside.

**Reference description**
A part-solid mass with major axis of 36 mm is found
in the periphery of the lower lobe S6/S9/S10 of the left lung.
It has well-defined and lobulated margins with pleural indentation.
Air-bronchogram and cavity are found inside. No calcification is found inside.

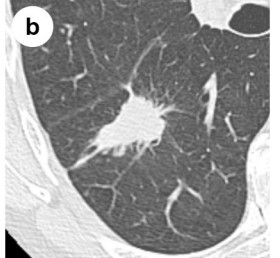

**Generated description**
An irregular, solid nodule with major axis of 29 mm is found
in the lower lobe S6 of the right lung.
It has lobulated and spiculated margins. Pleural indentation is found.
No calcification, cavity and air-bronchogram inside.

**Reference description**
A solid nodule with major axis of 29 mm is found
in the middle layer of the lower lobe S6 of the right lung.
It has spiculated margins with pleural indentation, and invades the upper lobe
S2 beyond the interlobar fissure.
Air-bronchogram is found inside. No calcification is found inside.

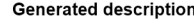
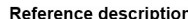

**Generated description**
An irregular, solid mass with major axis of 48 mm is found
in contact with pleura in the lower lobe S7/S8/S10 of the right lung.
It has lobulated and spiculated margins.
Air-bronchogram is found inside. No calcification inside.

**Reference description**
A solid mass with major axis of 48 mm is found
around the hilum of the lower lobe S7/S8/S9/S10 of the right lung.
It has spiculated margins with pleural indentation.
Air-bronchogram is found inside. No calcification and cavity inside.
It is in contact with the pleura on the diaphragm.

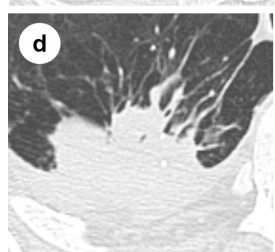

**Generated description**
An irregular, solid mass with major axis of 79 mm is found
in contact with pleura in the upper lobe S2/S3 of the right lung.
It has lobulated and spiculated margins.
Calcification is found inside. No cavity and air-bronchogram inside.

**Reference description**
A solid mass with major axis of 79 mm is found extending
from the hilum to the periphery of the upper lobe S1 and S2 of the right lung.
It has lobulated, lagged and spiculated margins.
Spot-like calcification is found inside.
It is in contact with the pleura widely.

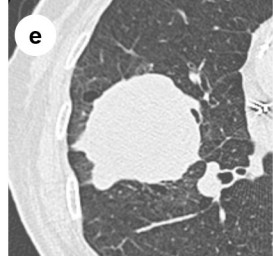

**Generated description**
An irregular, solid mass with major axis of 59 mm is found
in contact with pleura in the middle lobe S4 of the right lung.
It has lobulated and lagged margins.
No calcification, cavity and air-bronchogram inside.

**Reference description**
A solid mass with major axis of 59 mm is found
in the middle layer to the periphery of the middle lobe S4 of the right lung.
It has lobulated, lagged and spiculated margins with pleural indentation.
No calcification and cavity inside.
It is in contact with the pleura partially.

**Fig 9. Examples of cases including images and descriptions.** "Generated description" was generated by our system. "Reference description" was created by the co-authored radiologist from scratch (without our system). Both original sentences are written in Japanese and translated into English for explanation. The underlined word in generated description indicates the FP content of the generated description. The underlined word in reference description indicates the FN content of the generated description. Reprinted from https://www.cancerimagingarchive.net/ collection/rider-lung-ct/ under a CC BY license, with permission from Zhao, B., Schwartz, L. H., & Kris, M. G., original copyright 2015.

segments. We considered that the edge loss improved the segmentation accuracy by forcing the model to output the bronchopulmonary segment boundaries more naturally. We supposed that the slightly low exact match rate for both our method and radiologists was due to the large individual differences in the interpretation of the area of nodules with ill-defined margins and on the bronchial bifurcations in the lower lobes. Similar trends were observed in previous studies [30].

Many studies have been conducted to predict malignancy from nodule images using DL models trained with malignancy scores given by radiologists or definitive diagnosis results [31–33]. However, these methods may have limitations when it comes to assisting radiologists in managing lung nodules due to the potential lack of a sufficient basis for diagnosis. To manage lung nodules, providing basis for diagnosis is required: image characteristics. Previous studies have developed systems to recognize a small number of characteristics such as opacity and spiculation of lung nodules on CT images, but they did not encompass characteristics to assess the malignancy of lung nodules [34]. Here, we defined 14 types of characteristics to encompass the characteristics required for quantitative assessment of malignancy based on the lung cancer guidelines [2,7], Lung-RADS [21], and actual clinical practice. The evaluation experiment revealed that the model that input a wide-area image with an enlarged image together showed an improvement in AUC of over 3% in the boundary, edge, and cavity classes compared to the model that did not input an enlarged image. We considered that enlarged images helped the model recognize these classes effectively because it was necessary to capture the fine features of the edges and small low opacity areas inside to judge these classes. Furthermore, inputting an enlarged image together implicitly indicate the network location of the target nodule. This might also contribute to the improvement of the results.

Radiology reports were created and presented in the form of text by the radiologist to inform the client physician of the findings and diagnosis of the patient. Here, we adopted RNN-based sequence-to-sequence models instead of fixed phrases to generate lung nodule descriptions as natural Japanese sentences because the lung nodules show various characteristics, and it was considered that there were many combinations thereof, appearing impossible to prepare innumerable fixed phrases. Although previous studies have automatically generated findings for plain X-ray images [35], there have been no previous studies that applied to CT image reports that describe complex and detailed findings. In the experiment, we evaluated the description generation model on the two axes of adequacy and fluency. In the evaluation of adequacy, some cases omitted input contents in the description, although no cases incorrectly described un-entered contents in the description. In the evaluation of fluency, only one grammatical error occurred locally (particle error). We believe these performances were sufficient for practical use because we assumed that the lung nodule descriptions generated by our system should be confirmed and corrected by radiologists, as shown in the interpretation experiment.

Previous studies on the automatic generation of radiology reports [11,12] did not carry out interpretation experiments as ours did, so it was an underlying question whether such a system would be clinically useful. Through the interpretation experiment, we showed that, by using our system, the agreement of the contents described in the lung nodule descriptions of findings between two radiologists increased and their contents were more comprehensive without sacrificing precision. In short, we confirmed that our system was effective in uniformizing the lung nodule descriptions of high-quality findings. The results showed that our system assisted radiologists in correctly describing content, especially those related to the bronchopulmonary segments and pleura. The bronchopulmonary segment not only informed the reader of the nodule location, but also the arteries leading to the nodule, which is important for planning surgery. Our model classified pleural indentation signs with slightly lower accuracy than

radiologists; however, when used together, our model improved radiologist judges more comprehensively. As the pleural indentation sign is one of the risk factors for invasive pulmonary adenocarcinoma and an important finding [36], we believe that it has a great clinical significance in reducing the omission of such findings. However, we confirmed that our system induced false statements of findings in a few cases. For example, there were three cases in which our system mistakenly judged that there were no air-bronchogram signs, and the radiologists adopted this (This case is listed in Fig 9). However, in all of these cases, the air-bronchogram signs only appeared at the margins of the lesion, and it was possible that some radiologists might interpret these cases as the absence of air-bronchogram signs. In this experiment, the radiologists treated the lung nodule descriptions generated by our system as drafts and made additional corrections as necessary, so there were few cases where incorrect descriptions were left.

This study had some limitations. First, the interpretation experiment was small, and there were only two radiologists as subjects. The effect of the system might differ depending on the number of years of interpretation experience. In future work, we will examine the differences in the effects between inexperienced radiologists and experts. Second, we have not compared the results of nodule classification with the eventual pathological features. In future work, we plan to investigate the impact of this AI system on the differential diagnosis of lung nodules by comparing its output with the pathological features. Third, the study did not evaluate whether our system contributed to reducing the burden on radiologists. There is a chronic shortage of radiologists in Japan and the UK; hence, it is urgent to reduce their workload [37,38]. As our system could perform the process of identifying the name of the anatomical position where the target nodule was located, picking up characteristics, and creating drafts of descriptions in findings automatically, we expected that this system could reduce the workload of radiologists in clinical practice. In future work, we will compare the time required to write a report and the quality of the report between radiologists with our system and those without our system and evaluate whether the former achieved higher quality in a short time.

In conclusion, we developed a system that generated descriptions of findings from lung nodules on CT images. In the experiment, we showed that our system achieved similar accuracy as radiologists with a realistically collectable number of training datasets by developing the system in three modules: bronchopulmonary segment prediction, nodule classification, and description generation. Furthermore, we also showed that our system assisted radiologists in uniformizing the lung nodule descriptions of findings and in improving their quality. The workflow of summarizing the anatomical position and characteristics of a lesion in a description of findings was the same regardless of the organ and the type of lesion; therefore, our approach can be applied to other lesions, such as liver and kidney tumors.

## Supporting information

**S1 Fig. The structure of deep convolutional neural network for lung lobe/bronchopulmonary segments segmentation.** It consists of 22 3D-convolution layers, 3 max pooling layers, and 3 3D-deconvolution layers. Batch normalization layer and ReLU layer following each convolution layer.
(TIF)

**S2 Fig. The structure of deep convolutional neural network for lung nodule classification.** It consists of 12 3D-convolution layers and 2 fully connected layers. The output of the final convolution layer is compressed to 128 dimensions by global average pooling layer. Batch normalization layer and ReLU layer following each convolution layer. C represents the number of

classes to be classified, which we set to 16 in our experiment.
(TIF)

**S3 Fig. Thumbnails of the 32 nodule images used for the observer performance test.**
Reprinted from https://www.cancerimagingarchive.net/collection/rider-lung-ct/ under a CC
BY license, with permission from Zhao, B., Schwartz, L. H., & Kris, M. G., original copyright
2015.
(TIF)

**S1 Data. Thumbnails of the 32 nodule images used for the observer performance test.** Radiologist-generated descriptions and gold standards for the nodules, along with statistical data of
the observer performance test.
(XLSX)

## Acknowledgments

We would like to thank Editage (www.editage.com) for English language editing.

## Author Contributions

**Conceptualization:** Keigo Nakamura, Shingo Iwano, Shinji Naganawa.

**Data curation:** Yohei Momoki, Akimichi Ichinose.

**Formal analysis:** Yohei Momoki, Akimichi Ichinose.

**Investigation:** Yohei Momoki.

**Methodology:** Yohei Momoki, Akimichi Ichinose.

**Project administration:** Keigo Nakamura.

**Resources:** Keigo Nakamura, Shingo Iwano.

**Software:** Yohei Momoki, Akimichi Ichinose.

**Supervision:** Shinji Naganawa.

**Validation:** Yohei Momoki, Shingo Iwano, Shinichiro Kamiya, Keiichiro Yamada.

**Visualization:** Akimichi Ichinose.

**Writing – original draft:** Yohei Momoki, Akimichi Ichinose.

**Writing – review & editing:** Yohei Momoki, Akimichi Ichinose, Keigo Nakamura, Shingo
Iwano.

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
