## [Decision Letter · Decision Letter 0]

12 Dec 2023

PONE-D-23-25184Development of automatic generation system for lung nodule finding descriptionsPLOS ONE

Dear Dr. Momoki,

Thank you for submitting your manuscript to PLOS ONE. After careful consideration, we feel that it has merit but does not fully meet PLOS ONE’s publication criteria as it currently stands. Therefore, we invite you to submit a revised version of the manuscript that addresses the points raised during the review process.

The manuscript has been evaluated by two reviewers, and their comments are available below.

The reviewers have made a number of requests for additional detail or clarification, particularly with respect to methods and analyses. 

Could you please carefully revise the manuscript to address all comments raised?

We look forward to receiving your revised manuscript.

Kind regards,

Steve Zimmerman, PhD

Senior Editor, PLOS ONE

6. We note that Figure(s) 1, 2, 4, 6 and 9 in your submission contain copyrighted images. All PLOS content is published under the Creative Commons Attribution License (CC BY 4.0), which means that the manuscript, images, and Supporting Information files will be freely available online, and any third party is permitted to access, download, copy, distribute, and use these materials in any way, even commercially, with proper attribution. For more information, see our copyright guidelines: http://journals.plos.org/plosone/s/licenses-and-copyright.

a. You may seek permission from the original copyright holder of Figure(s) 1, 2, 4, 6 and 9 to publish the content specifically under the CC BY 4.0 license. 

Reviewers' comments:

Reviewer's Responses to Questions

**Comments to the Author**

1. Is the manuscript technically sound, and do the data support the conclusions?

Reviewer #1: Yes

Reviewer #2: Yes

2. Has the statistical analysis been performed appropriately and rigorously? 

Reviewer #1: Yes

Reviewer #2: Yes

3. Have the authors made all data underlying the findings in their manuscript fully available?

Reviewer #1: Yes

Reviewer #2: Yes

4. Is the manuscript presented in an intelligible fashion and written in standard English?

Reviewer #1: Yes

Reviewer #2: No

5. Review Comments to the Author

Reviewer #1: Dear Authors and Editors

I had the pleasure to review this very interesting manuscript by Momoki et al. on the development of automatic generation system to characterize and describe lung nodule according to radiological features.

The paper is overall very well-written even it is focused on a cross topic between radiology, physic, and bioengineering.

I have not found any major concerns or issues since it is a very thorough description of an innovative system designed with the aim of help the radiologists rather than replace them.

I have just some questions to the authors that should be solved.

1) Which were the inclusion criteria of all the radiology reports (known lung nodules, oncologic history, screening CT, etc.) ?

2) Regarding the bronchopulmonary segments’ prediction, have the author evaluated the system on patients with previous lung resection, with lesions involving two or more lobes or in cases of pulmonary atelectasis or Azygos lobe?

3) Have the author evaluated the presence of any anatomical features that could improve the match of the report as the presence of complete fissures.

4) Have the authors compared the description of the lung nodule with eventual pathological features?

5) In the same way, is it possible evaluate the system by adding the time evolution of the same lung nodule in order to better distinguish benign lesions from the malignant ones?

Reviewer #2: I am pleased to have the opportunity to review this paper.

I would congratulate the authors could develop novel and helpful automatic generation system for lung nodule finding descriptions.

The process of developing the system and its structure are described very precisely in the manuscript, and the efficacy of the system is likely to be evaluated in a proper manner.

However, I think the technical descriptions regarding the system construction, or the evaluation may be too complicated for general readers.

Therefore, I strongly request that they should be revised to be more understandable for many readers.

Additionally, I have several questions about the study.

In various deep learning process, on what basis was the sample size determined?

In line 262, the experiments were performed 5 times. In line 270, the experiments were performed 7 times. What is the difference between 2 sentences?

Other minor points of my concern are as follows.

Line 93 autors→authors

Please correct.

Regarding the slice thickness in table1, the scale is centimeter (cm).

Is it correct?

6. PLOS authors have the option to publish the peer review history of their article (what does this mean?). If published, this will include your full peer review and any attached files.

Reviewer #1: No

Reviewer #2: No

---

## [Author Response · Author response to Decision Letter 0]

16 Feb 2024

I have uploaded the letter as attachment labeled "Response to Reviewers", so please check it there.

---

## [Editor Report · Decision Letter 1]

27 Feb 2024

Development of automatic generation system for lung nodule finding descriptions

PONE-D-23-25184R1

Dear Dr. Momoki,

We’re pleased to inform you that your manuscript has been judged scientifically suitable for publication and will be formally accepted for publication once it meets all outstanding technical requirements.

Kind regards,

Vittorio Aprile

Guest Editor

PLOS ONE

Additional Editor Comments (optional):

Dear Author

I really appreciated the big effort made to modify the text according to the reviewer's advice.

---

## [Editor Report · Acceptance letter]

12 Mar 2024

PONE-D-23-25184R1 

PLOS ONE

Dear Dr. Momoki, 

I'm pleased to inform you that your manuscript has been deemed suitable for publication in PLOS ONE. Congratulations! Your manuscript is now being handed over to our production team.

Kind regards, 

on behalf of

Dr. Vittorio Aprile 

Guest Editor

PLOS ONE